**Title:** Development of an advanced, interpretable, and validated, machine learning platform, for the discovery of novel acetylcholinesterase inhibitors

**Authors:** Nirajan. Bhattarai[1], Marvin K. Schulte[1]*

Biomedical and Pharmaceutical Sciences, Skaggs College of Pharmacy, Kasiska Division of Health Sciences, Idaho State University, Pocatello, Idaho 83209, USA

The development of advanced algorithms, comprehensive exploration of molecular descriptor/feature spaces, model interpretability, and cross-species validation collectively facilitate rapid, accurate, and informed artificial intelligence-driven drug discovery. In this study, we specifically applied these approaches to the identification of novel acetylcholinesterase Inhibitors (AChEIs). Rapid identification of acetylcholinesterase inhibition can aid in the identification of new therapeutic agents as well as the identification of potential toxic effects of drug molecules. This study used four major Machine Learning (ML) variants—aggregated (ensemble + classical), AutoML (tpot), graph-based, and fine-tuned BERT large language model (LLM) ChemBERTA. All models were trained using a curated ChEMBL dataset of AChEIs, employing an IC50 activity cutoff of 1 uM for classification. Models underwent extensive cross-validation, training across 11 different feature sets, consistent hyperparameter optimization, and evaluation using six different species datasets including independent human dataset. Interpretation methods included identifying important features with LIME, fragment/atomic contributions for a prediction, and determining optimal thresholds/hyperparameters. Experimental evaluation of selected compounds will be carried out using modified Ellman's method. The study identified top performing models, including ensemble extra tree classifier with rdkit features (AUC ROC=0.92) among aggregrate, AutoML (tpot) extra tree classifier with rdkit features (AUC ROC=0.92), graph convolution network (AUC ROC=0.93) among graph based model, and fine-tuned ChemBERTA (ROC AUC=0.91). Extensive validation and hyperparameter optimization demonstrated the superiority of aggregate and AutoML (tpot) models in predicting AChEIs selective for human compared to other species. Promising hits were obtained through screening FDA-approved drugs, natural products from COCONUT, and ZINC's purchasable 250k compounds, followed by docking studies. The study showed a prediction threshold of 0.8 for optimal prediction of positives (>95% accuracy for positives) and pinpointed important features and key structural predictors. In conclusion, the study identified robust machine learning models for rapid detection of acetylcholinesterase inhibition, particularly rdkit feature-based aggregated and AutoML models, which could effectively discriminate human AChE inhibitors from other species.

**Keywords: Machine learning, drug discovery, acetylcholinesterase inhibitors, validation, interpretation, screening**

