# OpenReview forum: "Development of an advanced, interpretable, and validated, machine learning platform, for the discovery of novel acetylcholinesterase inhibitors"
_IEEE.org/ICIST/2024/Conference — IEEE ICIST 2024 Conference Submission_

### Official Review · Reviewer_YF9h · 2024-08-21
**This paper develops and validates an advanced, interpretable machine learning platform for discovering novel acetylcholinesterase inhibitors. However, the following suggestions need to be considered in the revised manuscript to further improve the quality of this paper.**

**Rating:** 7
**Confidence:** 3

**Review:**

1. How does this paper demonstrate the superiority of ensemble and AutoML (tpot) models in predicting AChE inhibitors?
2. What methods were used to ensure consistent optimization across different feature sets?
3. How robust are these models when applied to new compounds in practical settings? Are there specific challenges or limitations encountered?

---

### Official Review · Reviewer_jPxk · 2024-08-21
**Accept**

**Rating:** 7
**Confidence:** 4

**Review:**

This study has carried out a comprehensive and in-depth exploration of advanced machine learning techniques in the rapid and accurate identification of acetylcholinesterase inhibitors ( AChEIs ). By using a variety of cutting-edge methods, including ensemble methods, AutoML, graph-based models and fine-tuned large language models ( LLMs ), this study demonstrates the ability of AI-driven drug discovery. The robustness and reliability of the results were highlighted by rigorous cross-validation using the rigorously reviewed ChEMBL dataset and different species datasets. The reviewer has the following questions:

1. How well do the identified models generalize to other classes of inhibitors beyond acetylcholinesterase? Could the same approaches be adapted for different therapeutic targets with similar efficacy?

2. While the study emphasizes model interpretability using LIME and fragment contributions, how practical are these interpretability methods in a real-world clinical setting where quick and actionable insights are crucial?

3. The research shows a strong ability to predict human-specific AChEIs, but how might the models perform when faced with less well-characterized species or strains? Could this approach be extended to predict species-specific inhibitors for a broader range of organisms?

---

### Official Review · Reviewer_8e2a · 2024-08-21
**Good paper**

**Rating:** 7
**Confidence:** 3

**Review:**

This paper has developed an advanced, interpretable, and validated machine-learning platform for the discovery of novel acetylcholinesterase inhibitors. The paper is interesting and well-written. However, some questions need to be answered:
1.How did the integration of advanced algorithms and comprehensive exploration of molecular descriptor/feature spaces contribute to the success of the drug discovery process in identifying novel acetylcholinesterase inhibitors?

2.What are the key advantages of using ensemble and AutoML models over traditional machine learning approaches in predicting acetylcholinesterase inhibition, particularly in distinguishing human AChE inhibitors from those of other species?

3.How do the experimental validation and docking studies support the predictions made by the machine learning models, and what insights do they provide into the potential therapeutic and toxicological effects of the identified AChE inhibitors?

---

### Decision · Program_Chairs · 2024-09-08

Accept (Oral)